# Detection of Lysyl Oxidase Activity in Tumor Extracellular Matrix Using Peptide-Functionalized Gold Nanoprobes

**DOI:** 10.3390/cancers13184523

**Published:** 2021-09-08

**Authors:** Han Young Kim, Mihee Jo, Ju A La, Youngjin Choi, Eun Chul Cho, Su Hee Kim, Youngmee Jung, Kwangmeyung Kim, Ju Hee Ryu

**Affiliations:** 1Department of Biomedical-Chemical Engineering, The Catholic University of Korea, Bucheon 14662, Gyeonggi-do, Korea; hy0408@catholic.ac.kr; 2Center for Theragnosis, Biomedical Research Institute, Korea Institute of Science and Technology, Seoul 02792, Korea; jmh1327@naver.com (M.J.); yjchoi@kist.re.kr (Y.C.); kim@kist.re.kr (K.K.); 3Department of Chemical Engineering, Hanyang University, Seoul 04763, Korea; 024079@kist.re.kr (J.A.L.); enjoe@hanyang.ac.kr (E.C.C.); 4R&D Center, Medifab Ltd., Seoul 08584, Korea; sweess@imedifab.com; 5Center for Biomaterials, Biomedical Research Institute, Korea Institute of Science and Technology, Seoul 02792, Korea; winnie97@kist.re.kr

**Keywords:** lysyl oxidase, gold nanoparticles, colorimetric assays, extracellular matrix, tumor stiffness

## Abstract

**Simple Summary:**

Although various malignant tumors express high levels of lysyl oxidase (LOX) and though its role in tumor progression is well-defined, there is a lack of sensing techniques to target LOX. This study highlights the application of peptide-functionalized gold nanoprobes for sensing the LOX levels in tumor microenvironments. The gold nanoparticles (AuNPs) in these nanoprobes aggregate upon exposure to LOX, resulting in a red shift of the surface plasmon resonance peak, accompanied by a characteristic color change. This colorimetric assay based on peptide-functionalized AuNP sensitively detects LOX secreted from various cancer cells not only in vitro but also in the tissue extract. In this study, the suggested analytical approach demonstrated high specificity to LOX and did not show any color change in the presence of other enzymes.

**Abstract:**

High LOX levels in the tumor microenvironment causes the cross-linking of extracellular matrix components and increases the stiffness of tumor tissue. Thus, LOX plays an important role in tumorigenesis and in lowering the tumor response to anticancer drugs. Despite comprehensive efforts to identify the roles of LOX in the tumor microenvironment, sensitive and accurate detection methods have not yet been established. Here, we suggest the use of gold nanoparticles functionalized with LOX-sensitive peptides (LS-AuNPs) that aggregate upon exposure to LOX, resulting in a visual color change. LOX-sensitive peptides (LS-peptides) contain lysine residues that are converted to allysine in the presence of LOX, which is highly reactive and binds to adjacent allysine, resulting in the aggregation of the AuNPs. We demonstrated that the synthesized LS-AuNPs are capable of detecting LOX sensitively, specifically both in vitro and in the tissue extract. Moreover, the suggested LS-AuNP-based assay is more sensitive than commonly employed assays or commercially available kits. Therefore, the LS-AuNPs developed in this study can be used to detect LOX levels and can be further used to predict the stiffness or the anticancer drug resistance of the tumor.

## 1. Introduction

The extracellular matrix (ECM) is a highly dynamic complex that continuously undergoes the deposition, degradation, and remodeling [1]. The ECM comprises a diverse group of macromolecules, such as collagen, elastin, proteoglycans, non-collagenous glycoproteins, and elastic fibers [2]. Tissues are shaped by the bidirectional communication between the ECM and resident cells as well as by ECM remodeling [3]. ECM remodeling is crucial for tissue formation and the restoration of tissue homeostasis during wound repair [4]. However, tumors dysregulate and leverage ECM remodeling, which promotes tumorigenesis and metastasis [5]. The tumor stroma is characterized by the remodeling and stiffening of the ECM [6,7,8]. Abnormal ECM deposition and stiffness are observed during the development of cancer [9]. The stiffness of the ECM is heterogeneous throughout the tissues; however, it is well known that collagen is a major contributor to ECM stiffness [10]. The cross-linking of collagen and ECM stiffening accompanies tumor progression [6]. During the intramolecular or intermolecular cross-linking of collagen, the collagen fibers become increasingly insoluble and show progressive incremental stiffening [11,12]. Thus, the stiffness and elasticity of the tumor microenvironment have been shown to play a critical role in tumorigenesis.

Lysyl oxidase (LOX) is an ECM remodeling enzyme that is abundantly expressed in the tumor microenvironment [13,14]. In the ECM of tumor tissues, LOX induces the cross-linking of collagen, contributing to an increase in the ECM stiffness [15]. Specifically, LOX oxidatively deaminates specific lysine or hydroxyllysine residues, which form aldehyde groups, yielding allysine [16]. Through the enzymatic oxidation of the telopeptides of collagen by LOX, highly reactive aldehyde groups can react with the adjacent aldehyde groups, can render intramolecular cross-linking of collagen, or can react with other ε-amino groups of hydroxyllysine residues of an adjacent helix to render the intermolecular cross-linking of collagen [17]. Hydrogen peroxide is released during this oxidative reaction. Activated LOX increases the ECM stiffness, regulates cell migration, and promotes cancer malignancy [18]. Indeed, high-LOX levels have been correlated with ECM stiffness and poor prognosis in breast, colorectal, head and neck, and prostate cancer [19,20,21]. Furthermore, several studies have suggested that the increase in the ECM stiffness caused by high-LOX levels may also confer resistance to chemotherapy [22]. A rigid ECM acts as a protective barrier, which occludes anticancer agents from penetrating the tumor cells [23]. In this regard, recent studies have reported that LOX inhibition suppresses lung metastasis [24] and further provides a pre-clinical rationale for using LOX inhibitors to potentiate chemotherapy responses in terminal cancers such as pancreatic cancer [25] and triple negative breast cancer [26]. Together, the sensitive determination of LOX levels in tumors may enable the prediction of the stiffness of tumor ECM and thereby may provide more accurate and precise ways to enhance anticancer effectiveness in patients with cancer. However, despite the comprehensive research that defines the roles of LOX in tumor progression, a diagnostic platform for the sensitive detection of LOX activity in heterogeneous tumors has not yet been established.

In the past decade, gold nanoparticle (AuNP)-based colorimetric sensors have attracted considerable interest as a rapid and simple sensing tool [27,28]. Due to plasmon resonance, stabilized AuNPs can present a characteristic extinction peak at a specific wavelength. This extinction peak can be shifted toward longer wavelengths upon the aggregation of AuNPs, which results in the AuNP suspension undergoing a color change, turning from a mostly pinkish red to blue/purple [29]. This color change can be used to detect the presence or amounts of analytes that cause the selective aggregation of AuNPs [30,31]. Owing to their unique optical properties, controllable size, and catalytic properties, AuNP-based sensors have been used in numerous innovative approaches [32]. Furthermore, the versatile surface chemistry of AuNPs facilitates the conjugation of various molecular probes for the detection of biological and chemical targets such as metal ions [33], small molecules [34], proteins [35], and nucleic acids [36]. Importantly, compared to other sensing tools, a AuNP-based colorimetric sensor is a simple method for the sensitive detection of the target because the color change can be directly monitored without sophisticated instruments. Therefore, medical and analytical approaches using AuNPs to detect various molecules in organs or tissues, especially in tumors, are promising, owing to their simplicity, which reduces the time and cost of the procedure. 

In this study, we utilized LOX-sensitive hexapeptides (LS-peptides; Figure 1a) to functionalize AuNPs as a colorimetric sensor to detect LOX both in vitro and in tumor tissue extract. LS-peptide-functionalized AuNPs (LS-AuNPs) aggregate by interacting with LOX, thereby yielding a plasmonic color change (Figure 1b). The color and extinction spectra of LS-AuNPs in solution were monitored after the addition of conditioned media collected from LOX-high- or LOX-low-expressing cancer cells (Figure 1c). Moreover, we show the potential use of LS-AuNPs for the detection of LOX levels in tumor tissues isolated from mice tumor models and further examine the correlation between the LOX levels and the collagen content and the ECM stiffness in the tumor.

## 2. Materials and Methods

### 2.1. Materials

LOX enzymes were purchased from OriGene (Rockville, MD, USA). Matrix metalloproteinase-2 (MMP-2) and cathepsin B enzymes were purchased from R&D Systems, Inc. (Minneapolis, MN, USA). The LOX inhibitor, β-aminopropionitrile, was purchased from Sigma-Aldrich (St. Louis, MO, USA). Fetal bovine serum (FBS) was purchased from Invitrogen Canada (Burlington, ON, Canada), and RPMI 1640 medium was obtained from Gibco BRL (Gaithersburg, MD, USA). For Western blot analysis, anti-LOX and a horseradish peroxidase-conjugated anti-mouse antibody were purchased from Aviva Systems Biology (San Diego, CA, USA) and Santa Cruz Biotechnology (Santa Cruz, CA, USA), respectively. A commercially available LOX assay kit, Amplite^TM^ fluorimetric LOX Assay kit, was purchased from AAT Bioquest (Sunnyvale, CA, USA). HAuCl_4_ and tri-sodium citrate were purchased from Sigma-Aldrich and Fisher Scientific (Waltham, MA, USA), respectively.

### 2.2. Cell Culture

Human breast cancer cell lines, MCF-7 and MDA-MB-231, were purchased from the Korean Cell Line Bank (KCLB, Seoul, Korea) and were cultured in RPMI 1640 medium containing 10% (*v*/*v*) fetal bovine serum (FBS), penicillin G (100 U/mL), and streptomycin (100 μg/mL). MCF-7 (approximately at 1 × 10^6^ cells/dish) and MDA-MB-231 cells (approximately at 1 × 10^6^ cells/dish) were maintained on a 100 mm tissue culture dish and were incubated at 37 °C with 5% CO_2_.The culture medium was exchanged every three days. For the evaluation of LOX in culture medium, the MCF-7 and MDA-MB-231 cells were cultured for 24, 48, or 72 h and the LOX-containing conditioned media were collected.

### 2.3. Synthesis of AuNP

AuNPs were prepared according to a citrate reduction procedure [37]. A solution of HAuCl_4_ (95 mL, 0.26 mM) was heated and equilibrated at 100 °C in a reactor. After the equilibrium, tri-sodium citrate (0.5% (*w*/*v)*) dissolved in deionized water (DW, 5 mL) was added to the reactor. After the synthesis of the AuNPs at 100 °C for 30 min, the suspension containing the AuNPs was cooled to room temperature.

### 2.4. Preparation of LS-AuNP and Control AuNP

LS-peptides (Ala-Ala-Lys-Ala-Ala-Cys) and LOX-insensitive peptides (control peptides, Ala-Ala-Ala-Ala-Ala-Cys) were synthesized using a standard solid-phase peptide synthesis method (Peptron, Daejeon, Korea) [38]. Peptide-functionalized AuNPs (1.3 nM) were prepared by mixing freshly prepared AuNPs in DW (1 mL) with an aqueous solution of LS-peptides or control peptides (5 mg/mL, 0.1 mL), yielding LS-AuNPs or control AuNPs, respectively. After reacting at room temperature for 1 h, the peptide-functionalized AuNPs were purified by centrifugation, washed twice with DW, and re-dispersed in DW (1.3 nM). The dispersed LS-AuNPs and control AuNPs were stored at 4 °C until further use.

### 2.5. Characterization of LS-Peptides and LS-AuNP

LS- or control peptides treated with the LOX enzyme were characterized by analytical reverse phase-high performance liquid chromatography (RP-HPLC): 20% to 80% acetonitrile containing 0.1% trifluoroacetic acid (TFA) versus DW containing 0.1% TFA over 30 min at a flow rate of 1.0 mL/min. Mass spectrometry was used to measure the molecular weight of the LS- or control peptides treated with the LOX enzyme using Varian 500-MS (Varian Inc., Palo Alto, CA, USA). The mean diameter and size distribution of the AuNPs and LS-AuNPs were observed using dynamic light scattering (DLS) at 25 °C. The UV/Vis absorbance of AuNPs or the peptide-labeled AuNPs with or without various enzymes was recorded from 400 to 700 nm using a UV/Vis spectrophotometer (Optizen 2120, Mecasys, Daejeon, Korea). AuNP aggregates were observed using transmission electron microscopy (TEM, CM30 electron microscope, Philips, CA, USA) operating at 80 kV. The sample solution was placed on the grid for 2 min, and excess solution was blotted using filter paper. For staining, the grid was placed on a drop of 2% (*w*/*v*) uranyl acetate.

### 2.6. Sensitivity and Specificity of Peptide-Functionalized AuNPs for LOX Detection

The sensitivity of the LS-AuNPs or control AuNPs was examined by incubating the LS-AuNPs (1.3 nM) in DW containing various concentrations (6.0, 12.0, 24.0, 48.0, and 96.0 nM) of LOX enzymes. Color changes were observed, and the UV/Vis absorbance of the LS-AuNPs or control AuNPs was measured. The specificity of the LS-AuNPs was examined by incubating the LS-AuNPs (1.3 nM) in DW containing 15 nM of activated MMP-2, cathepsin B, and LOX plus β-aminopropionitrile (LOX inhibitor). The color change was monitored, and the UV/Vis absorbance of the LS-AuNPs were measured.

### 2.7. LOX Assays

A commercially available LOX activity assay kit, the Amplite^TM^ fluorimetric LOX kit, was used to validate the LS-AuNP-based results. This assay utilizes a proprietary LOX substrate that can release hydrogen peroxide when reacted with LOX. In this assay, LOX activity was indirectly detected by measuring the production of hydrogen peroxide. In practice, conditioned media (50 µL) was added to the assay reaction mixture (50 µL) in a 96-well plate. After incubating for 30 min at 37 °C, the signal was read at 576 nm using an absorbance plate reader.

### 2.8. In Vivo Tumor Model and Preparation of Tumor Lysates

All of the animal experimental procedures were performed in compliance with the institutional guidelines of the Korea Institute of Science and Technology and the relevant laws. MCF-7 (approximately 1 × 10^6^ cells/mouse) and MDA-MB-231 cells (approximately 1 × 10^7^ cells/mouse) suspended in phosphate-buffered saline (PBS, 100 μL) were subcutaneously injected into the left flank of athymic nude mice (20 g, Orient, Seoul, Korea). The LOX inhibitor (3 mg/kg body weight) was intraperitoneally injected daily for 14 days after the injection of the tumor cells. When the tumors reached 7.0 ± 0.5 mm in diameter, they were excised for further analysis. For protein extraction, the excised tumor tissues were ground in liquid nitrogen, rinsed with PBS, and sonicated in lysis buffer (7 M urea, 2 M thiourea, 4% (*v*/*v*) CHAPS, 130 mM dithiothreitol, 1 mM NaF, Na_2_VO_3_, and a complete protein inhibitor mixture) for 20 min. The samples were then centrifuged at 9700× *g* for 15 min. The protein concentration in the supernatant was measured using a Bradford assay (Bio-Rad protein assay kit; Bio-Rad, Hercules, CA, USA).

### 2.9. Histological and Western Blot Analyses

Excised tumor tissues were fixed in 4% (*v*/*v*) buffered formalin, dehydrated with a graded ethanol series, and embedded in paraffin. The specimens were cut into 5 μm-thick sections and were stained using the Masson′s trichrome (MT) method to detect collagen. For immunohistochemistry (IHC) analysis, sections were stained with primary antibodies against LOX. The staining signals were developed with a Histostain^®^-Plus Kit (Invitrogen, Burlington, ON, Canada). The nuclei were stained with 4′,6-diamidino-2-phenylindole (DAPI). 

For Western blot analysis, each aliquot of the cell supernatant or extracted proteins were mixed with a sample buffer (0.25 M of Tris, 0.8% (*w*/*v*) sodium dodecyl sulfate, 10% (*v*/*v*) glycerol, 0.05% (*w*/*v*) bromophenol blue, pH 6.8) and were run on a 10% polyacrylamide gel after boiling for 10 min. The gels were transferred to a blot membrane using iBLOT (Invitrogen, Burlington, ON, Canada). Membranes were blocked with milk in Tris-buffered saline-Tween20 (TBST) at room temperature for 1 h and were then incubated with anti-LOX at 4 °C overnight. A total of three washes (15 min each) with TBST were performed, and the membranes were incubated with a horseradish peroxidase-conjugated anti-mouse antibody for 1 h in TBST and were then washed three times for 15 min each with TBST. β-Actin was used as an internal control. Immunoreactive proteins were visualized by enhanced chemiluminescence.

### 2.10. Compressive Modulus Measurements

Compression testing was performed on cylinidrical shape-cutting tumor tissues using an electro-mechanical driven indenter (Instron 5966) including a force transducer (a load cell of 10 N), a stepper motor, and a linear displacement transducer [39]. The tangent elastic moduli of the tumor tissues were calculated from the initial linear slope in the resulting stress–strain curves. 

### 2.11. Statistical Analysis

The data acquired in this study were presented as mean ± standard deviation (SD). Values are representative of three independent experiments with three or more samples per group. The *p*-values were calculated by one-way analysis of variance (ANOVA) with Tukey′s post hoc test. Graphs and plots were generated using GraphPad Prism software. Statistical significance was set at *p* < 0.05.

## 3. Results

### 3.1. Reactivity of LS-Peptides and Control Peptides to LOX

LS-peptides, which are hexapeptides consisting of six amino acid residues containing lysine (Ala-Ala-Lys-Ala-Ala-Cys), were used in this study for the detection of LOX. Control peptides, which are hexapeptides without lysine (Ala-Ala-Ala-Ala-Ala-Cys), were used as controls. LOX is known to deaminate lysine to allysine, which is highly reactive. The aldehyde groups in allysine can react with the adjacent aldehyde groups or with the ε-amino group of a lysine residue [40]. Thus, we hypothesized that the LOX peptide may form a secondary structure in the presence of LOX. We first evaluated the response of hexapeptides to LOX using RP-HPLC (Figure 2a). The LS-peptides showed a distinct response to LOX, compared to the control peptides, which did not respond to LOX. As demonstrated by the HPLC profiles, the major peak of the LS-peptide showed a clear shift in the retention time after the addition of LOX (from ~17 min to ~26 min). In contrast, the addition of LOX did not result in a peak shift of the control peptides. Reponses of the hexapeptides to LOX were further evaluated by mass spectroscopy (Figure 2b). The LS-peptides showed a peak at 575.8 Da in the product before the addition of LOX. After the addition of LOX, it showed an additional peak at 1151.3 Da, indicating conjugation between the two LS-peptides. However, the control peptides did not show any change in its molecular weight after the addition of LOX. These results clearly demonstrated that the LS-peptides become active upon the addition of LOX and interact with adjacent hexapeptides, yielding a secondary structure.

### 3.2. Synthesis and Characterization of LS-AuNPs

As we demonstrated that the addition of LOX activates the LS-peptides, we next prepared AuNPs functionalized with the LS-peptides (LS-AuNPs). AuNPs coated with control peptides (control AuNPs) served as a control. The hexapeptides used in this study were designed to contain cysteine on one side of the peptide, and the thiol groups in cysteine were expected to strongly interact with the AuNPs [33]. First, as-synthesized AuNPs and LS-AuNPs were observed by TEM (Figure 3a). TEM observation exhibited that both the as-synthesized AuNPs and the LS-AuNPs were spherical in shape, with a minor increase seen in the size of the LS-AuNPs. For clarification, the sizes of the AuNPs and the LS-AuNPs were further evaluated by DLS measurements (Figure 3b and Table 1). The hydrodynamic diameter of the AuNPs increased from 14.76 ± 0.38 nm in diameter to 24.76 ± 0.29 nm in diameter, indicating the presence of LS-peptides on the surface of the AuNPs. In addition, from UV/Vis spectroscopy (Figure 3c), the localized surface plasmon resonance peak of the AuNPs slightly shifted from 518 nm to 522 nm after the surface modification of the AuNPs with the LS-peptides. We confirmed the conjugation of the LS-peptides to the AuNPs via the measurement of their zeta potentials (Table 1). The as-synthesized AuNPs showed a zeta potential of −30.0 ± 5.6 mV, which is consistent with a previous study that reported the synthesis of AuNPs based on a standard citrate reduction process [41]. After the surface modifications, the negative charge of the AuNPs became positive (21.5 ± 0.6 mV for LS-AuNPs), which was likely due the conjugation of the LS-peptides, which had a theoretical isoelectric point of 8.27. 

### 3.3. Aggregation and Color Change of LS-AuNPs by Addition of LOX

We next evaluated whether the synthesized LS-AuNPs aggregate and show a color change after the addition of LOX. First, a series of LOX concentrations (0–96 nM) were added to the LS-AuNPs. The color of the LS-AuNP suspension and its UV/Vis extinction were monitored after incubation for 1 h, and TEM observations were conducted. The suspension exhibited a stepwise color change from the original pinkish red to a purple color (Figure 4a). In the TEM images, the LS-AuNPs were found to be well dispersed in the solution before the addition of LOX, whereas it was found to form aggregates in the presence of 96 nM of LOX (Figure 4b). The extinction spectra of the LS-AuNPs underwent red-shift and broadened with increasing LOX concentration (Figure 4c). Next, quantitative analysis was performed by measuring the extinction values at 650 nm (E_650_) normalized by extinction values at 520 nm (E_520_; Figure 4d). The values of E_650_/E_520_ showed a linear range (R^2^ = 0.99) from 6 to 96 nM. Meanwhile, we did not observe any noticeable changes in the color or the UV/Vis spectra of the LS-AuNPs when other enzymes, including MMP-2 or cathepsin B, were added to the suspension of LS-AuNPs (Figure 4e). Importantly, the pretreatment with β-aminopropionitrile (LOX inhibitor) prevented changes in the UV/Vis spectra or color of the LS-AuNPs upon exposure to LOX. We also examined the changes in the color or the UV/Vis spectra of the control AuNPs after the addition of increasing concentrations of LOX (0–96 nM), and no significant difference was observed (Figure 4f). These results collectively demonstrated that LS-AuNPs aggregates upon exposure to LOX, thereby changing their colors, and that the responses are highly specific to LOX compared to other enzymes. Furthermore, we demonstrated that the presence of lysine in the amino acid residues of hexapeptides is critical for the detection of LOX, as indicated through the comparison with the control AuNPs. 

### 3.4. Detection of LOX in Conditioned Media of Cancer Cells Using LS-AuNPs

We investigated the potential use of LS-AuNPs to determine the levels of LOX released from various cancer cells. In this study, MDA-MB-231 cells (high LOX expression) and MCF-7 cells (low LOX expression) were selected for a comparative study using LS-AuNPs [42]. LOX is a secreted enzyme; hence, LOX detection was performed in the conditioned media of these cancer cells. We first compared the color change of the solution containing LS-AuNPs after the addition of conditioned media collected from either MDA-MB-231 or MCF-7 cells (Figure 5a). When the conditioned media from a 72-h culture of MDA-MB-231 cells were added to the LS-AuNP solution, the color changes and spectral shifts (Figure 5b) were more significant compared to those from the addition of the conditioned media of a 72-h culture of MCF-7 cells. The calculation of the E_650_/E_520_ values (Figure 5c) showed that there was a significant difference in these values between the two LS-AuNP suspensions treated with the conditioned media of either MDA-MB-231 or MCF-7 cells. In addition, as the culturing period of the MDA-MB-231 cells increased from 24 h to 72 h, the color changes and spectral shifts increased, possibly due to the accumulation of the LOX released from the MDA-MB-231 cells in a time-dependent manner. 

Next, LOX detection using Western blot analysis or using a commercially available LOX assay kit was performed to validate the accuracy of LS-AuNPs for the detection of LOX and to compare their sensitivity. Interestingly, the LOX in the conditioned media collected from the 72-h culture of MDA-MB-231 cells was barely detectable by Western blot analysis (Figure 5d). The LOX content in the conditioned media collected from the 24-h and 48-h culture of MDA-MB-231 and from the 72-h culture of MCF-7 cells was not detected by Western blot analysis. A commercially available LOX assay, the Amplite^TM^ fluorimetric LOX assay, is a fluorescence signal-based method for detecting the hydrogen peroxide released in a reaction between a LOX substrate and LOX [43]. Importantly, LOX activity from the commercially available assay revealed a very similar tendency; however, it showed lower sensitivity. (Figure 5e). In contrast to the results of the LS-AuNP-based measurement of LOX (Figure 5c), there was no significant difference in LOX activity between the conditioned media of the 24-h and 48-h cultures of MDA-MB-231 cells. These results highlight the accuracy and high sensitivity of LS-AuNP-based sensing in comparison with the common assays used for the detection of LOX levels, such as Western blotting or fluorimetric assays.

### 3.5. Detection of LOX Levels in Tumor Tissues of Mice Models

We next determined the feasibility of using LS-AuNPs for the detection of LOX in the extracted tissues. Mouse tumor models were established by the subcutaneous inoculation of MDA-MB-231 and MCF-7 cancer cells. To examine the effect of the LOX inhibitor in vivo, the LOX inhibitor (3 mg/kg) was intraperitoneally injected every day until 14 days after the inoculation of MDA-MB-231 cells. When tumors reached 7.0 ± 0.5 mm in diameter, tumor lysates of mice with MDA-MB-231 tumors, MDA-MB-231 tumors treated with LOX inhibitor, and MCF-7 tumors were collected and treated with LS-AuNPs (Figure 6a). Western blot analysis confirmed higher levels of LOX expression in the MDA-MB-231 tumor microenvironment compared to those of the MDA-MB-231 tumors treated with the LOX inhibitor or the MCF-7 tumors (Figure 6b). Furthermore, IHC staining of tumor tissue sections against LOX revealed strong LOX expression in the MDA-MB-231 tumors and limited LOX expression in the MCF-7 tumors (Figure 6c). In addition, reduced LOX expression was observed in the MDA-MB-231 tumors treated with the LOX inhibitor. Next, the amount of collagen tumor tissue was evaluated using Masson′s trichrome (MT) staining. Collagen in the fibrils was stained blue. Higher amounts of the fibrillar collagen present within the MDA-MB-231 tumors appeared blue compared to the MCF-7 tumors, which is likely due to high levels of LOX causing collagen cross-linking. The MDA-MB-231 tumors of the LOX inhibitor-injected mice lost collagen-containing fibrils. Furthermore, compression testing revealed the incremental stiffening of the MBA-MB-231 tumors compared to the MCF-7 tumors (Figure 6d). The elastic modulus of the MCF-7 tumors (52 ± 12 kPa) was 11% lower than that of the MBA-MB-231 tumors (457 ± 171 kPa). The LOX-inhibitor-treated MBA-MB-231 tumors (402 ± 121 kPa) had a slightly lower value than the MBA-MB-231 tumors without the LOX inhibitor treatment. As we evaluated the high levels of LOX that partially contributed to increased collagen-containing fibrils and the stiffness in the MDA-MB-231 tumors, we next added tumor lysates prepared from the respective tumor tissues to the LS-AuNPs to determine whether the LS-AuNPs could sensitively detect the LOX levels in the tumor microenvironment. Distinct color changes (Figure 6e) and spectral shifts in the plasmon band (Figure 6f) were observed in the LS-AuNPs treated with the tumor lysates of MDA-MB-231 tumors in comparison with the MCF-7 or MDA-MB-231 tumors treated with LOX inhibitors.

## 4. Discussion

A significant amount of experimental evidence supports the notion that changes in LOX levels alter ECM mechanics and contribute to cancer progression, and several studies have demonstrated that LOX activity is closely correlated with tumor stiffness and the invasiveness of cancer [6,20,44]. In this study, AuNPs functionalized with LOX-sensitive peptides were introduced to detect the LOX levels in both in vitro and tumor tissue extracts. It was assumed that the hexapeptides that were changed to a reactive state by LOX can bind to adjacent molecules, which can induce the aggregation of AuNPs via the abridging of neighboring AuNPs. The spectral shift and color changes in the solution containing AuNPs are well known to occur because of the changes in the aggregate size, aggregate dimensions (e.g., linear or three-dimensional), and interparticle spacing [45,46]. In this study, the extinction spectra underwent red shift and broadened with increasing LOX concentrations. In addition, an obvious color change was observed from pinkish red to purple. Further supporting evidence of the aggregation of the LS-AuNPs was gathered by TEM observation. TEM images showed that the LS-AuNPs were significantly associated/assembled after the addition of LOX. Together, these results indicate that LOX induces the aggregation and subsequent spectral shift and color change of AuNPs in suspension. 

The LOX levels can be directly observed by the color change of the AuNP suspensions with the “naked eye,” or can be quantified with extinction spectral measurements using the LS-AuNP-based assay. In the present study, quantitative analysis using LS-AuNPs showed a detection limit in the nanomolar range of the LOX concentration. This may be further improved by inserting an increased number of lysine residues in the LOX-sensitive peptides; however, various factors such as solubility, isoelectric point, and peptide length should be considered to facilitate the surface coating of AuNPs and to enhance their reactivity against LOX. We compared the sensitivity of our AuNP-based LOX detection method to common detection methods, which included Western blot analysis or commercially available LOX assay. The commercially available LOX assay indirectly detects the production of the hydrogen peroxide that is released from the reaction between the LOX substrate and LOX. Thus, commercially available LOX assay kits may not specifically detect LOX, especially at the tissue level, which is a major drawback of this method. It is not surprising that hydrogen peroxide can be generated by multiple enzymes and in multiple cellular compartments in cancer tissues [47,48]. The aberrant generation of hydrogen peroxide is correlated with the development and progression of cancer [47,49]. Colorimetric analytical approaches for the detection of other cancer biomarkers (e.g., carcinoembryonic antigen, p53 protein, or glucose oxidase) using antibody- or DNA-conjugated AuNPs have been previously presented [50,51,52]. In spite of these efforts, currently, there is a lack of a colorimetric AuNP sensor that sensitively detects ECM remodeling enzymes such as LOX. Moreover, the high molecular weight of antibodies and the poor biostability of oligonucleotides are of great concern, and it is necessary to optimize the bioconjugation process to maintain the stability of conjugated AuNPs. Peptides with a precise sequence structure have a lower molecular weight compared to antibodies and show higher biostability than the oligonucleotides. With the rapid development of peptide synthetic technology, purposefully designed peptides that greatly reduce the complexity of research are commercially available. Thus, peptide-conjugated AuNPs facilitate a simple and sensitive assessment. Furthermore, because peptide library screening technology provides the facile development of peptide sequences that recognize a wide range of target molecules [53], peptide-conjugated AuNPs may serve as versatile diagnostic biosensors. Recently, Liu et al. introduced gold nanocluters integrated with protease-cleavable peptides for a visual readout of disease in a mouse tumor model [54]. Cleavable peptides may serve as a potential tool to regulate the aggregation and color change of gold nanoprobes; however, it is extremely challenging to design peptide sequences that can be selectively cleaved by only one target enzyme. In our study, color changes did not occur after the addition of other enzymes such as MMP-2 or cathepsin B, indicating that our analytical approach is specific to LOX detection. Indeed, these comparative studies should be further examined using other matrix remodeling enzymes that are highly expressed in tumors [55].

Active LOX is known to increase collagen cross-linking, which subsequently increases ECM stiffness. The LOX levels, which were measured using LS-AuNPs in various tumor tissue extracts, correlated directly with their collagen contents. This was evaluated by MT staining and ECM stiffness measured with a tensile loading assay. The measured stiffness in this study is correlated with previously reported invasive cancer properties; MDA-MB-231 cells are known as a highly invasive/metastatic breast cancer cell line, while MCF-7 cells are a poorly invasive/nonmetastatic breast cancer cell line [56,57]. We demonstrated that high LOX-expressing MDA-MB-231 tumors contain larger amounts of fibrillar collagens and a higher elastic modulus than low LOX-expressing MCF-7 tumors. Additionally, these fibrillar collagens were reduced by treatment with the LOX inhibitor, β-aminopropionitrile. This is consistent with previous findings [6,43]; however, we did not observe a statistically significant difference in the elastic modulus of MDA-MB-231 tumors after treatment with the LOX inhibitor. Although the LOX inhibitors hindered collagen cross-linking and lowered the ECM stiffness in vivo, various other factors are involved in ECM remodeling. For example, α-smooth muscle actin (α-SMA), growth factors, MMPs, and integrins have been reported to regulate ECM stiffness [44]. Thus, the simultaneous inhibition of these molecules can be further tested to effectively reduce tumor stiffness. Interestingly, MDA-MB-231 cells are triple-negative breast cancer cells, and their chemoresistance is a major obstacle for successful treatment [26]. It is well known that altered and hardened ECM induced by LOX is one of the causes of chemoresistance [58]. Therefore, the rapid and sensitive detection of LOX in tumor tissue extracts may provide information on whether the tumor is susceptible or resistant to anticancer drugs. Moreover, a quick and accurate determination of LOX levels in the tissue specimen may facilitate the early diagnosis of cancer. For the clinical translation of the suggested approach, a fine-needle aspiration biopsy can be conducted, and the determination of cancer biomarkers in the collected specimen can be achieved by simply incubating the sample with plasmonic nanoparticles [59]. Thus, the suggested AuNP-based approach provides a great opportunity for cancer diagnosis and personalized cancer therapy.

## 5. Conclusions

LOX, an ECM-remodeling enzyme, is known to strongly affect ECM stiffness via collagen cross-linking. AuNPs functionalized with LOX-sensitive hexapeptides were introduced to detect LOX levels as a colorimetric sensor in vitro as well as in tumor tissue extracts. We demonstrated that LS-AuNPs are capable of the sensitive detection of LOX after the direct addition of LOX and in cancer cells with different levels of LOX secretion. Importantly, the LS-AuNPs showed high LOX specificity and did not detect levels of other ECM enzymes, namely MMP-2 and cathepsin B. Moreover, we showed that the presence of a lysine residue in the hexapeptide is crucial for the color changes in solutions containing AuNPs, as evaluated by experiments with the control peptides and control AuNPs. This study revealed that LS-AuNP-based measurement is more sensitive than common detection assays such as Western blotting or fluorimetric assays. When the LS-AuNPs were applied to various tumor tissue extracts, the LOX levels correlated directly with their collagen content and ECM stiffness. The collagen content and the ECM stiffness were measured using MT staining and a tensile loading assay, respectively. Together, our data showed that the suggested AuNP-based approach is useful for LOX-specific detection with high sensitivity and can be used as an alternative tool to predict the ECM stiffness and drug responsiveness of the target tumors.

## Figures and Tables

**Figure 1 cancers-13-04523-f001:**
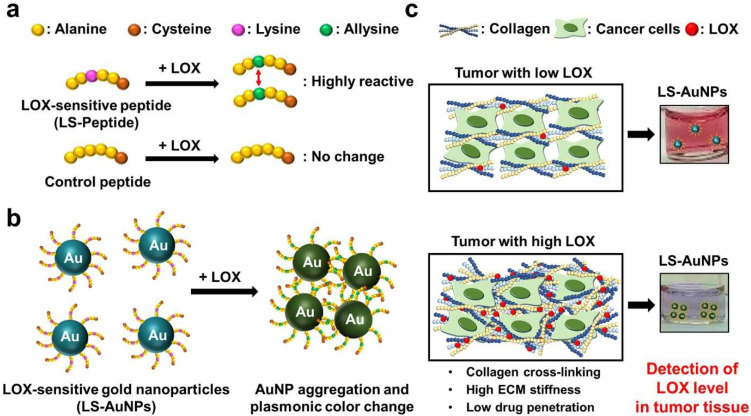
(**a**) The reaction between the lysyl oxidase (LOX)-sensitive hexapeptides containing lysine (LS-peptides) or hexapeptides without lysine (control peptides) and LOX. The addition of LOX caused lysine to convert to highly reactive allysine. (**b**) A schematic showing the principles of the colorimetric assay for LOX detection using LS-AuNPs. Due to the interactions between the highly reactive allysines, LS-AuNPs aggregate, thereby yielding a plasmonic color change. (**c**) The color of the AuNP suspension turns from the original pinkish red colour to blue/purple due to the aggregation of LS-AuNPs induced by the high LOX level in the tumor microenvironment.

**Figure 2 cancers-13-04523-f002:**
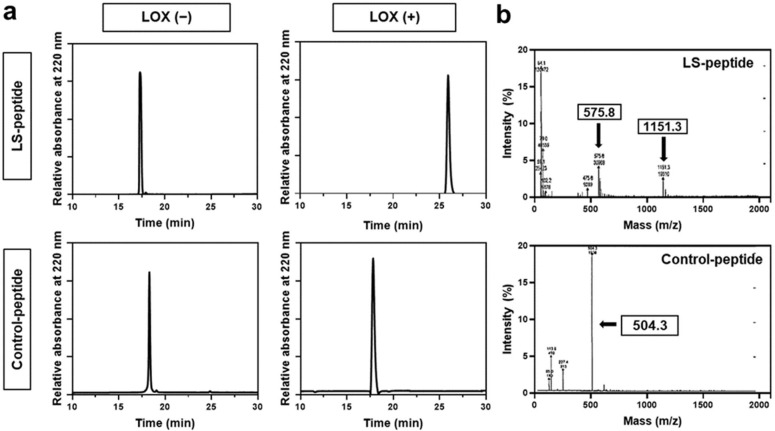
(**a**) HPLC profiles of the LS-peptides and control peptides before and after the addition of LOX. (**b**) Mass spectrometry of the reaction mixture LS-peptides + LOX or control peptides + LOX.

**Figure 3 cancers-13-04523-f003:**
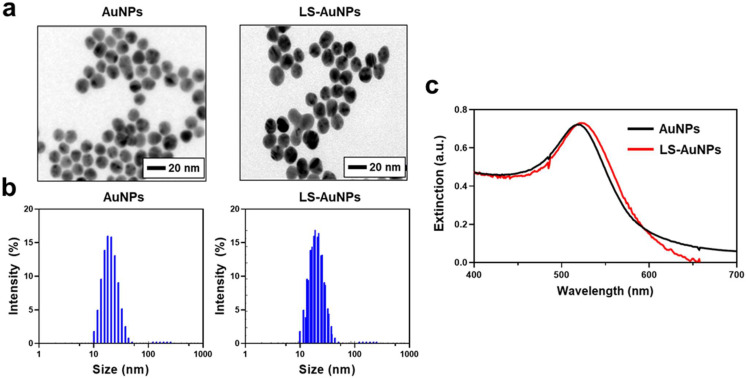
(**a**) Transmission electron microscopy (TEM) images of as-synthesized AuNPs and LS-AuNPs. (**b**) Size distribution of AuNPs and LS-AuNPs, as measured by dynamic light scattering (DLS). (**c**) UV/Vis extinction spectra of AuNPs and LS-AuNPs.

**Figure 4 cancers-13-04523-f004:**
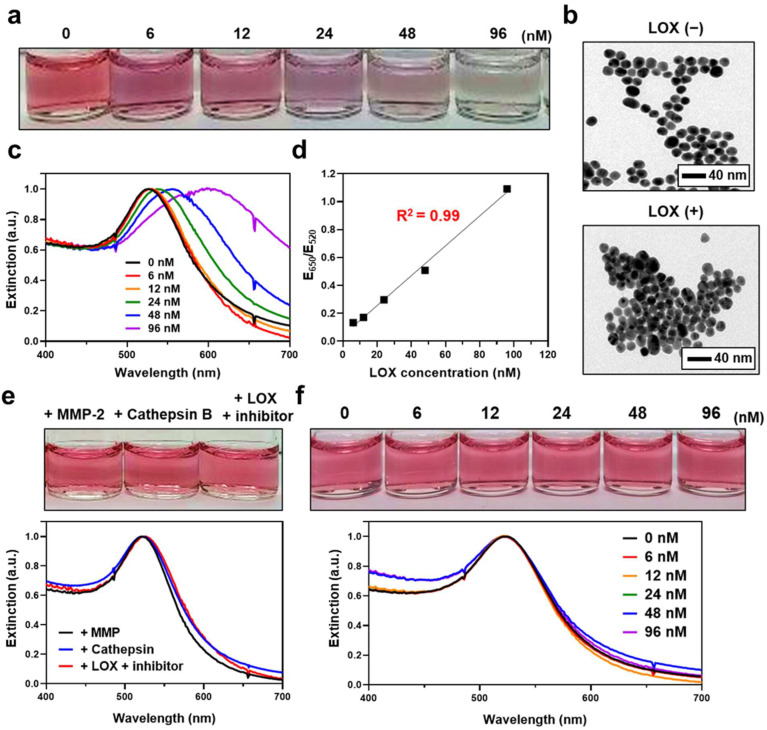
(**a**) Photographs showing the colors of LS-AuNP suspension after the addition of different concentrations of LOX. (**b**) TEM images of the LS-AuNPs before (upper panel) and after (lower panel) the addition of LOX (96 nM). (**c**) Changes in the UV/Vis extinction spectra of LS-AuNPs with different concentrations of LOX. (**d**) Calibration curve for LOX determination. Quantitative analysis was performed by measuring the extinction values at 650 nm normalized by extinction values at 520 nm (E_650_/E_520_). (**e**) Photographs and corresponding UV/Vis extinction spectra of LS-AuNPs after the addition of MMP-2, cathepsin B, or LOX + LOX inhibitor (β-aminopropionitrile). (**f**) Photographs and corresponding UV/Vis absorption spectra of control AuNPs after the addition of different concentrations of LOX.

**Figure 5 cancers-13-04523-f005:**
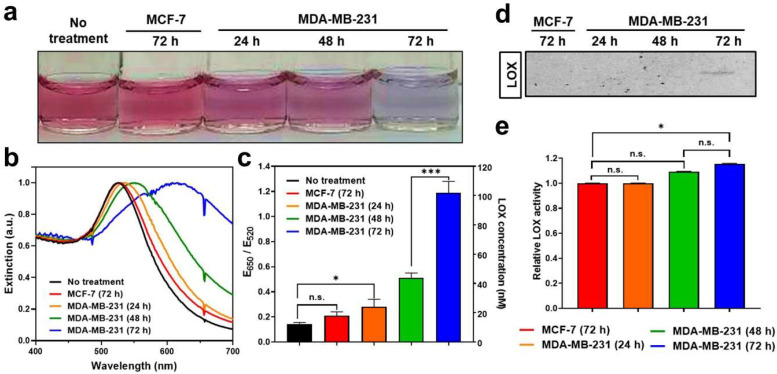
(**a**) Photographs of LS-AuNP suspension before or after the addition of conditioned media collected from a 72-h culture of MCF-7 cells; 24-h, 48-h, and 72-h cultures of MDA-MB-231 cells; (**b**) corresponding UV/Vis absorption spectra; and (**c**) E_650_/E_520_ values. (**d**) Western blot analysis of LOX proteins contained in conditioned media collected from culture incubated for different periods. Original Western Blot figure can be found in Appendix A. (**e**) LOX detection in different conditioned media using a commercially available LOX assay. n.s. indicates not significant (*p* > 0.05), whereas the asterisk indicates significant difference (* *p* < 0.05 and *** *p* < 0.001).

**Figure 6 cancers-13-04523-f006:**
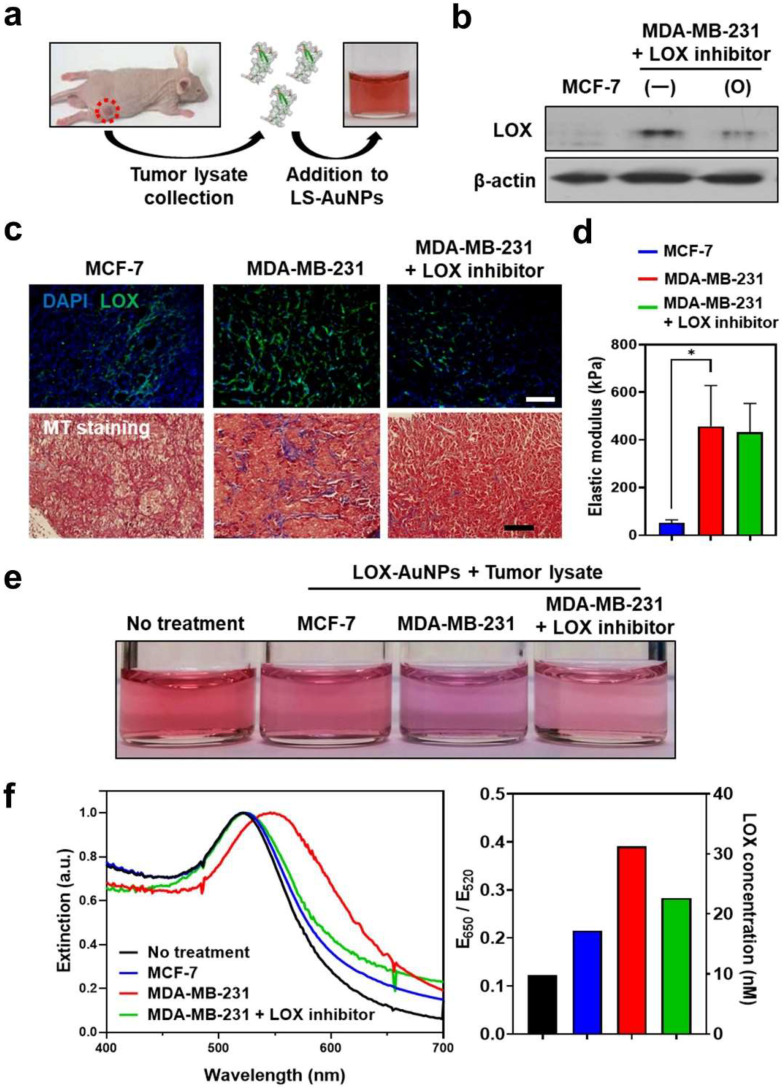
(**a**) Schematic illustration of the addition of tumor lysates collected from MCF-7 (low LOX expression) or MDA-MB-231 (high LOX expression) tumor-bearing mice to LS-AuNPs for the detection of the LOX level in tumor tissues. (**b**) Western blot analysis for the evaluation of LOX protein in tumor tissues of MCF-7- and MDA-MB-231-bearing mice with or without treatment with LOX inhibitor. Original Western Blot figure can be found in Appendix A. (**c**) IHC staining of LOX (upper panel) and Masson′s trichrome (MT; lower panel) staining of tumor tissues from different experimental groups. (**d**) Measurement of elastic modulus to compare the stiffness of different tumors. Asterisk indicates significant difference (* *p* < 0.05). (**e**) Photographs of LS-AuNP suspension before or after the addition of the tumor lysates collected from different animal groups. (**f**) UV/Vis extinction spectra (left panel) and E_650_/E_520_ value (right panel) of the LS-AuNPs suspension treated with various tumor lysates.

**Table 1 cancers-13-04523-t001:** Hydrodynamic diameters and zeta potentials of the AuNPs and LS-AuNPs (*n* = 3, mean ± SD)**.**

Nanoparticle	Hydrodynamic Diameter (nm)	Zeta Potential (mV)
AuNPs	14.76 ± 0.38	−30.0 ± 5.6
LS-AuNPs	24.76 ± 0.29	21.5 ± 0.6

## Data Availability

The data presented in this study are available on request from the corresponding author.

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
