# Peer review of "Detection of Lysyl Oxidase Activity in Tumor Extracellular Matrix Using Peptide-Functionalized Gold Nanoprobes"

_cancers, 2021, doi:10.3390/cancers13184523_

Round 1
Reviewer 1 Report
The following manuscript presents a quantitative method for analysis of LOX concentration both in vitro and in vivo for cancer detection. The research design is logical and appropriate. Several characterisation methods are well described. Literature review is full and relevant. However, the novelty of research is very questionable. Several similar methods based on gold nanoparticles/clusters interaction with cancer markers were presented before (e.g. M. Stevens group). I strongly recommend to highlight and explain novelty of the presented method, and add more details why is it more beneficial in comparison with alternatives. Also quality of figures and graphs should be improved significantly.
Author Response
"Please see the attachment."

Reviewer 2 Report
I congratulate the authors for such a nice work, including high-quality performance of experiments, presentation of outcomes/results, as well as clear writing. I recommend this manuscript for the publication after minor revision, as follows:
- Figure 5c and 6F (right panel): Add another Y axis for the LOX concentration corresponding to E650/E520 values based on the calculation shown in Figure 4d.
- This approach based on plasmonic nanoparticles has a potential to improve many aspects of cancer diagnostic technologies and assist pathologists. Therefore, I strongly recommend that authors briefly discuss its clinical translation and possible utilization for cancer diagnosis based on "fine-needle aspiration biopsy and plasmonic nanoparticles".
Author Response
"Please see the attachment."

Reviewer 3 Report
The authors presented an interesting tool to determine levels of LOX in tumors. However, claims presented in the abstract, introduction, and conclusions are not fully supported by the experimental evidence provided. My two major concerns are:
(1) Applicability of LOX-AuNPs for determination of the enzyme levels "at tissue level". This claim is present in many places in the manuscript: “to detect the LOX levels in both in vitro and tumor tissues” (discussion), “not only in vitro, but also at the tissue level.” (Summary), “LOX-AuNPs are capable of detecting LOX sensitively and specifically, both in vitro and at the tissue levels.” (Abstract), “to detect LOX levels as a colorimetric sensor in vitro, as well as in tumor tissues.” (conclusions). This is a misleading claim. Authors only confirmed their method at tissue extracts, which is not equal to detection IN tissues.
(2) sensitivity of the methods cannot be compared based on the samples of unknown composition (conditioned media). To present better sensitivity of LOX-AuNPs in comparison to other methods, I suggest performing control experiments on well-defined LOX solutions.
Minor comments:
Fig 1.a is not self-explaining, hard to follow
I would suggest replacing “LOX-peptide” with some other term to be more distinguishable from LOX itself. Similarly, LOX-AuNPs suggest conjugation with LOX rather than LOX-peptide.
Fig. 4 a – it seems that the concentration decreases. Is it only a matter of aggregation or something more? Please comment on that.
Fig. 6 F – statistics missing
Line 424: Claim that “The ECM stiffness is closely related to cancer progression and metastasis.” – should be described as broader or softened.
Line 440: Statement “It is well known that altered and hardened ECM induced by LOX plays a major role in the promotion of chemoresistance [53].” Is not true. Chemoresistance is a complex, multifactor phenomenon.
The overall article would also benefit if authors provided some ideas on how they foresee the nanoparticles' application in clinical/research use.
Author Response
"Please see the attachment."

Round 2
Reviewer 1 Report
Authors changed the manuscript accordigly all reviewer comments. The article is significantly improved and can be published in Cancers.